# A minimal biochemical route towards de novo formation of synthetic phospholipid membranes

Ahanjit Bhattacharya[1], Roberto J. Brea [1], Henrike Niederholtmeyer [1] & Neal K. Devaraj [1]

All living cells consist of membrane compartments, which are mainly composed of phospholipids. Phospholipid synthesis is catalyzed by membrane-bound enzymes, which themselves require pre-existing membranes for function. Thus, the principle of membrane continuity creates a paradox when considering how the first biochemical membrane-synthesis machinery arose and has hampered efforts to develop simplified pathways for membrane generation in synthetic cells. Here, we develop a high-yielding strategy for de novo formation and growth of phospholipid membranes by repurposing a soluble enzyme FadD10 to form fatty acyl adenylates that react with amine-functionalized lysolipids to form phospholipids. Continuous supply of fresh precursors needed for lipid synthesis enables the growth of vesicles encapsulating FadD10. Using a minimal transcription/translation system, phospholipid vesicles are generated de novo in the presence of DNA encoding FadD10. Our findings suggest that alternate chemistries can produce and maintain synthetic phospholipid membranes and provides a strategy for generating membrane-based materials.

[1] Department of Chemistry and Biochemistry, University of California, 9500 Gilman Drive, Natural Sciences Building 3328, San Diego, CA 92093, USA. Correspondence and requests for materials should be addressed to N.K.D. (email: ndevaraj@ucsd.edu)

Phospholipids are the primary constituents of cell membranes. In living organisms, phospholipids are generated enzymatically by the reaction of a polar head group with long-chain acyl derivatives. These key steps rely on integral membrane proteins, such as acyltransferases, which require pre-existing membranes for proper folding and function[1] (Fig. 1a). This mechanism implies that all biological membranes must arise from pre-existing membranes[2]. However, the principle of biological membrane continuity presents a challenge for explaining how phospholipid membranes were generated de novo before the current membrane-dependent enzymes and mechanisms for phospholipid synthesis developed. We reasoned that designing a minimal route for enzymatic de novo phospholipid synthesis could help us understand how early cellular membrane synthesizing machinery evolved[3–5]. Since present-day integral membrane proteins cannot carry out true de novo phospholipid formation, a method by which a soluble enzyme could facilitate the synthesis of membrane-forming phospholipids is required. It can also provide simplified strategies for generating membrane compartments in synthetic cells[6–9], enable the development of tools for reconstituting membrane proteins[10,11], and facilitate strategies for synthesizing structured lipids[12].

As there are no analogous reactions in biology, we sought to design a unique lipid synthesizing system by repurposing a soluble mycobacterial ligase, FadD10[13] for phospholipid formation (Fig. 1b). FadD10 catalyzes the generation of fatty acyl adenylates (FAAs) from fatty acid, $Mg^{2+}$, and ATP precursors. Acyl adenylates (AAs)— metabolic intermediates found in diverse biochemical pathways in both prokaryotes and eukaryotes, typically undergo enzymatic coupling with thiol nucleophiles, such as coenzyme A (CoA)[14]. AAs and other acyl phosphates can also react non-enzymatically with primary amines in aqueous media[15,16]. Therefore, we hypothesized that an FAA could spontaneously react with an amine-functionalized lipid fragment to produce a membrane-forming phospholipid (Fig. 1c).

Here, we show that FadD10 can mediate the de novo generation of phospholipid molecules from water-soluble single-chain amphiphilic precursors. The FAAs generated by FadD10 react chemoselectively with amine-functionalized lysolipids to form phospholipids that self-assemble to form membranes. This demonstrates that pathways radically different from those taking place in living cells may be developed for synthesizing membrane-forming materials.

## Results
**Reactivity of fatty acyl adenylates (FAA).** In order to evaluate the scope and applicability of the proposed lipid synthesis pathway, we first explored the reactivity patterns of FAAs. We synthesized dodecanoyl-AMP 1[17] (Supplementary Figs. 1–4) as a model FAA, and found that it was fairly stable to hydrolysis at 37 °C in 4-(2-hydroxyethyl)-1-piperazineethanesulfonic acid (HEPES) buffer, pH 7.5 in the absence or presence (10 mM) of $Mg^{2+}$, and over time scales relevant to our subsequent experiments (Supplementary Fig. 5a). We observed that 1 showed negligible reactivity toward the hydroxy groups of the naturally occurring lysolipids 1-oleoyl-2-hydroxy-*sn*-glycero-3-phosphocholine (Lyso $C_{18:1}$ PC-OH) and 1-palmitoyl-2-hydroxy-*sn*-glycero-3-phosphocholine (Lyso $C_{16:0}$ PC-OH) (Supplementary Fig. 5b). Remarkably, when we mixed 1 with the corresponding amine-functionalized lysolipids 2 (Fig. 1c and Supplementary Figs. 1–4) or 4 (Fig. 1c and Supplementary Figs. 1–4) in HEPES buffer, pH 7.5 at 37 °C, the solution became turbid after 20 min, and we observed the formation of a large population of vesicles (Supplementary Fig. 6 and Supplementary Movie 1). The progress of the reaction between 1 and 2 was analyzed over time using liquid chromatography–mass spectrometry (LC–MS) measurements (Fig. 1d). The second-order rate constant for the reaction was determined to be $87.0 \pm 9.1\ M^{-1}\cdot s^{-1}$ (Supplementary Fig. 5c, error value represents standard deviation from $n = 3$ replicates). We obtained the nuclear magnetic resonance (NMR) spectra from purified phospholipids 3 and 5. These data verified that acylation occurred at the primary amine functionality of 2 and 4, respectively.

Because amines are ubiquitous in biology, we asked if FAAs have any selectivity for amine-functionalized lysolipids over simple water-soluble amines. When the reaction between 1 and 2 was conducted in the presence of 50 mM lysine in HEPES buffer, we found that coupling occurred with excellent selectivity between the FAA and the lysolipid. We did not observe significant side products that corresponded to a reaction with the primary amine groups in lysine (Fig. 1e and Supplementary Fig. 5d). We found similar selectivity when the coupling reaction between 1 and 2 was carried out in 100 mM 2-amino-2-hydroxymethyl-propane-1,3-diol (Tris) buffer, pH 8.0 (Supplementary Fig. 5e). We followed the reaction kinetics between 1 and a non-amphiphilic primary amine, Fmoc-L-Lys-OH, by HPLC-MS. The second order rate constant for the reaction was obtained to be $0.0033 \pm 0.0004\ M^{-1}\cdot s^{-1}$ (error value represents standard deviation from $n = 3$ replicates), almost 30,000-fold slower than the reaction with the lysolipid 2 (Supplementary Fig. 5f). We attribute this high selectivity of phospholipid synthesis to hydrophobic interactions between the alkyl chains of 1 and 2, which are brought into close proximity, likely within mixed micelles or formed lipid membranes. These reactivity patterns indicate that FAAs could serve as an activated acyl precursor in a more complex biochemically relevant media.

**De novo phospholipid formation mediated by FadD10.** We next aimed to identify an appropriate enzyme capable of generating FAAs, such as 1 from fatty acid and ATP precursors. In biological systems, fatty acids are activated to FAAs by various fatty acyl CoA ligases (FACLs) and fatty acyl adenylate ligases (FAALs)[14]. While FACLs are common, the FAA intermediate remains tightly bound to the active site to facilitate its subsequent reaction with CoA, which limited their attractiveness as candidate enzymes. We sought an enzyme that could form FAAs that could subsequently react with amine-functionalized lysolipids. We selected FadD10[18], a recently characterized FAAL involved in the biosynthesis of a putative lipopeptide virulence factor in *Mycobacterium tuberculosis*[19]. FadD10 is a soluble enzyme that converts long-chain saturated fatty acids (typically $C_{12}$–$C_{16}$) into corresponding FAAs in presence of ATP and $Mg^{2+}$. Since FadD10 displays an "open" conformation of its active site, it does not have a high binding affinity to the FAA product[18]. As such, we hypothesized the newly synthesized FAAs will be free to diffuse away and react with amine-functionalized lysolipids to form the corresponding amidophospholipids.

We expressed N-terminal $His_6$-tagged FadD10 in *E. coli* and purified it according to a published procedure[18]. In a typical phospholipid synthesis reaction, FadD10 was incubated with ATP, $MgCl_2$, lysolipid 2, and sodium dodecanoate in HEPES buffer at 37 °C. The enzymatically formed FAA intermediate 1 was detected by HPLC-MS, even after FadD10 was separated from the reaction mixture by spin filtration (Supplementary Fig. 7a, b). These results indicate that 1 was released from the enzyme's active site. Approximately 30 min after mixing FadD10 and substrates, small vesicles could be detected by optical microscopy. These vesicles gradually transformed into larger vesicles of various sizes and lamellarity (Fig. 2a, Supplementary Fig. 8). We followed the progress of de novo formation of phospholipid membranes using a well-established Fluorescence

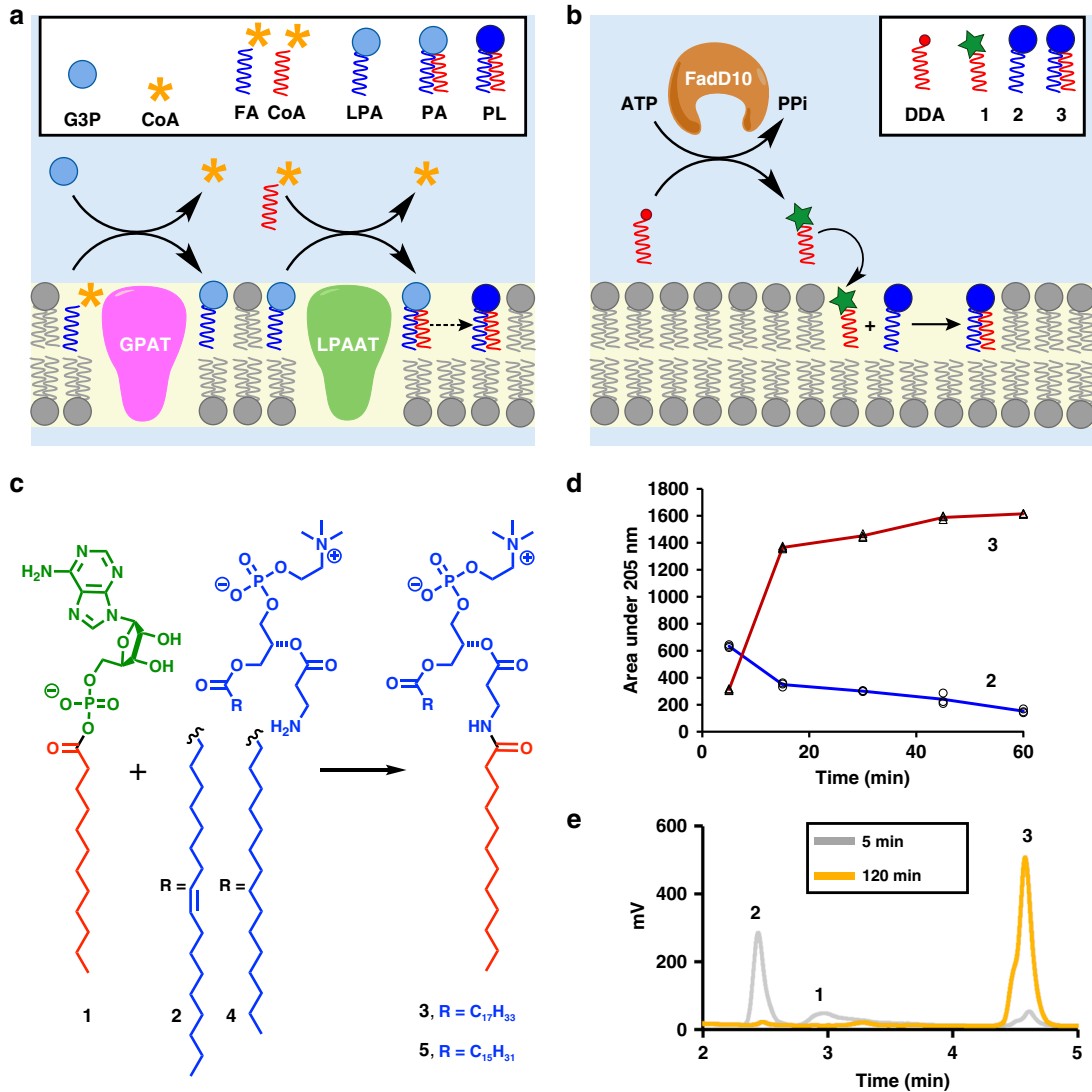

**Fig. 1** De novo formation of phospholipid membranes based on adenylate chemistry. **a** A representative phospholipid biosynthetic pathway (Kennedy pathway), which involves multiple membrane-bound enzyme-catalyzed steps, substrates and cofactors [GPAT, glycerol-3-phosphate acyl transferase; LPAAT, lysophosphatidic acid acyl transferase; G3P, glycerol 3-phosphate; CoA, coenzyme A; FACoA, fatty acyl coenzyme A; LPA, lysophosphatidic acid; PA, phosphatidic acid; PL, phospholipid]. **b** The proposed synthetic pathway of phospholipids, which involves a single soluble enzyme FadD10 and reactive lipid precursors [DDA, dodecanoic acid]. **c** De novo synthesis of phospholipids (**3** or **5**) by chemoselective reaction of the model FAA dodecanoyl-AMP (**1**) and amine-functionalized lysolipids (**2** or **4**). **d** Kinetics of phospholipid **3** (open black triangles, red line) formation by the reaction of FAA **1** with lysolipid **2** (open black circles, blue line). Integrated HPLC peak areas (205 nm) from three experiments were used to monitor the progress of the reaction. **e** HPLC/ELSD traces monitoring the selective formation of phospholipid **3** by reaction of FAA **1** and lysolipid **2** in the presence of 50 mM lysine

Resonance Energy Transfer (FRET) assay involving fluorescently labeled phospholipid probes[20]. We observed a linear increase in the ratio of the fluorescence intensities of the donor dye NBD-DHPE ($\lambda_{ex}$: 430 nm, $\lambda_{em}$: 530 nm) and the acceptor dye Rhodamine-DHPE ($\lambda_{ex}$: 430 nm, $\lambda_{em}$: 586 nm), suggesting that membrane growth is taking place during phospholipid synthesis (Supplementary Fig. 9). In a control experiment, where GTP was substituted for ATP, a nearly constant ratio of the donor and acceptor fluorescence intensities was observed, consistent with the lack of phospholipid formation.

In addition, we performed several control experiments where one of the key reaction components was omitted or replaced by an unreactive substitute to confirm that vesicle formation occurred due to phospholipid generation (Supplementary Fig. 10a–c). Cryo-electron microscopy further verified the presence of membranes (Fig. 2b). We analyzed the formation of phospholipid **3** using HPLC-MS and found that the precursors

were almost completely consumed in about 8 h (Fig. 2c). In order to test the broader applicability of our method, we carried out the reaction in presence of the amine-functionalized lysophosphatidylglycerol **6** (Supplementary Figs. 1–3). We obtained the bilayer forming phospholipid **7**, which self-assembled to form vesicles as well (Supplementary Fig. 11).

Interestingly, when we used Alexa Fluor® 488-labeled FadD10, we observed that the enzyme was significantly associated with the de novo-formed phospholipid **3** membranes (Fig. 2d and Supplementary Fig. 12a). We isolated the vesicles from the reaction mixture by spin filtration and subsequent SDS–PAGE analysis of the vesicle fraction showed the presence of FadD10 (Supplementary Fig. 12b). Further investigation using unilamellar vesicles encapsulating fluorescently labeled FadD10 and ATP showed that the enzyme gradually associated with the membranes when dodecanoic acid and lysolipid **4** were slowly supplied from the outside (Supplementary Fig. 12c, d and Supplementary

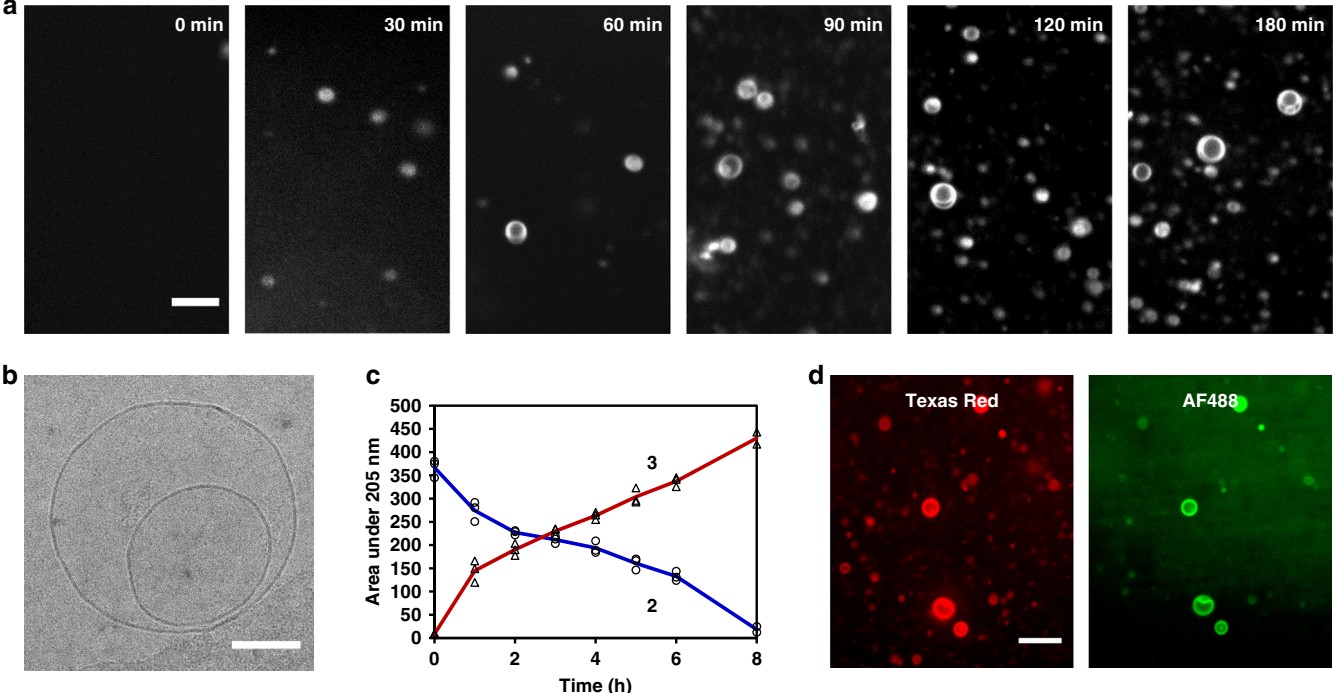

**Fig. 2** FadD10 mediated de novo formation and assembly of phospholipid membranes. **a** Time series of spinning disk confocal microscopy images depicting de novo phospholipid **3** vesicle formation resulting from the incubation of an aqueous solution of dodecanoic acid, lysolipid **2**, ATP, MgCl₂, FadD10, and 0.1 mol% Texas Red® DHPE at 37 °C. Scale bar: 10 µm. **b** Cryogenic-transmission electron microscopy (cryo-TEM) image of a de novo formed phospholipid vesicle showing the presence of membranes. Scale bar: 100 nm. **c** Kinetics of the consumption of lysolipid **2** (open black circles, blue line) and formation of phospholipid **3** (open black triangles, red line) at 37 °C. Integrated HPLC peak areas (205 nm) from three experiments were used to monitor the progress of the reaction. **d** Spinning disk confocal microscopy images of phospholipid **3** vesicles formed with Alexa Fluor® 488-labeled FadD10 and 0.1 mol% Texas Red® DHPE, showing association of the enzyme with the membrane. Scale bar: 10 µm

Movie 2). These data suggest that FadD10 associates with the membranes possibly through electrostatic interaction with the charged amphiphilic precursors. Given that the theoretical isoelectric point (pI) of His₆-FadD10 is 5.4 (ExPASy), it is likely to bear a negative charge in the pH range of our experiments and undergo electrostatic interactions with the positively charged amine-functionalized lysolipid to become encapsulated during de novo formation of phospholipid vesicles.

**Microfluidics experiments**. Since FadD10 could facilitate the de novo formation of phospholipids, we determined if vesicles encapsulating FadD10 could generate additional phospholipids if we continuously supplied reactive precursors. We expected to observe membrane growth and the formation of new vesicles occurring if lipid synthesis is efficient. Since continual feeding of substrates requires maintaining non-equilibrium steady-state conditions for extended periods of time, we utilized a microfluidic chip[21] (Fig. 3a) to trap giant vesicles (Fig. 3b; see Supplementary Fig. 13a–f for full characterization) composed of phospholipid **3** encapsulating Alexa Fluor® 488-labeled FadD10 and ATP. We then continuously flowed reactive precursors (lysolipid **4**, sodium dodecanoate, ATP, and MgCl₂ in HEPES buffer) for 12 h with simultaneous imaging (Fig. 3a). In all experiments, we carefully matched the osmolarities of the initial vesicle dispersion and the flow solution using an osmometer in order to minimize non-specific effects on membrane morphology[22,23] arising from osmotic mismatch. During the first few hours, the vesicles underwent drastic morphological transformations followed by growth (Fig. 3c and Supplementary Fig. 14a–c and Supplementary Movies 3 and 4). Some vesicles exhibited membrane growth that promoted division into smaller vesicles, while maintaining

their internal content, as observed using the fluorescence signal from labeled FadD10 (Fig. 3d and Supplementary Fig. 15a, b and Supplementary Movies 5 and 6). This mode of division resembles that in L-form bacteria, where excess membrane synthesis increases the cellular surface-area-to-volume ratio to produce proliferation and budding events[24]. The addition of amphiphilic precursors likely enhanced the permeability of the membranes to polar solutes such as ATP due to formation of local non-lamellar phases[25] or transient defects[26–28] similar to what has been observed in previous works.

In a subsequent experiment, we replaced the amine-functionalized lysolipid **4** with Lyso C₁₆:₀ PC-OH, a lysolipid unreactive toward **1** (n = 3; Supplementary Fig. 5b and Supplementary Movie 7). During the first 5 h, the vesicles underwent minor structural deformations without significant changes in size. The vesicles then started to solubilize likely due to surfactant action[29,30]. The chambers were depleted of vesicles over the next 3 h. In a separate experiment (described in Supplementary Methods), we simulated the conditions in the microfluidic device by slowly adding reactive precursors (lysolipid **4**, sodium dodecanoate, ATP, and MgCl₂ in HEPES buffer) to the giant vesicles encapsulating FadD10 and ATP using a syringe pump and observed the formation of new phospholipid **5** over time by HPLC-MS (Fig. 3e). These results demonstrate that the vesicle growth and division events we observed in the microfluidic device were the result of the synthesis of new phospholipid, and not simply due to incorporation of amphiphilic precursors to the membranes.

**Linking gene expression to phospholipid membrane formation**. Functional synthetic cells require coupled gene expression

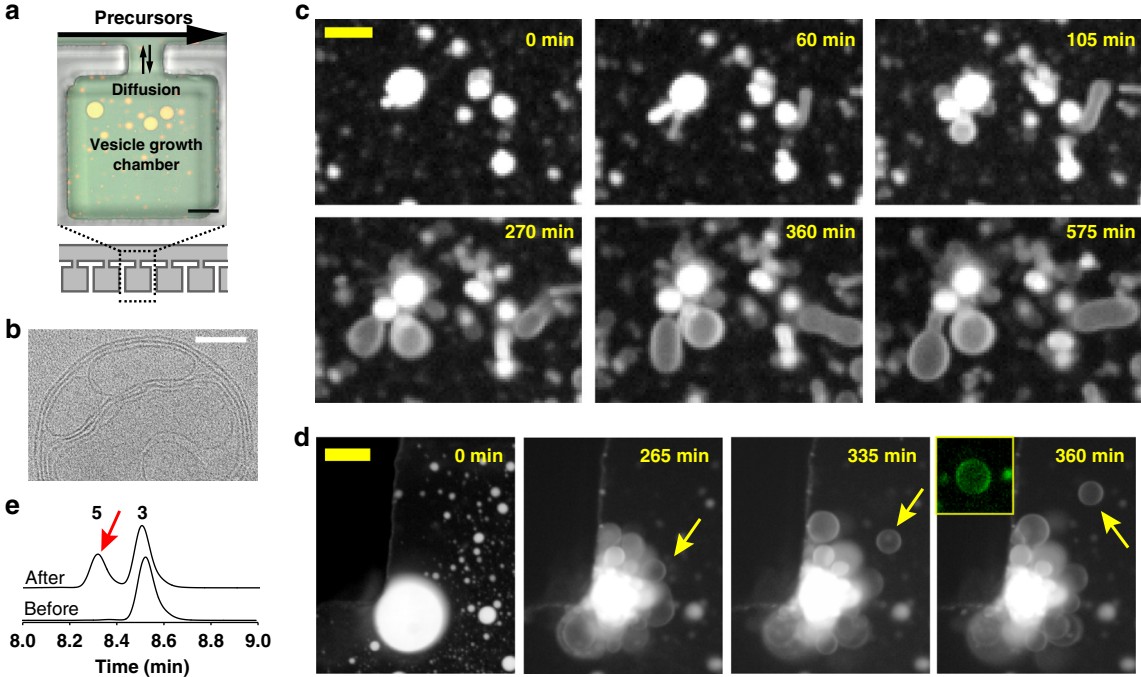

**Fig. 3** Membrane growth and division in a microfluidic device. **a** Schematic representation of a microfluidic chip utilized to entrap phospholipid **3** giant vesicles encapsulating Alexa Fluor® 488-labeled FadD10 and ATP. Reactive precursors (dodecanoic acid, lysolipid **4**, ATP, and MgCl₂) were continuously flowed with simultaneous spinning disk confocal microscopy. Scale bar: 50 μm. **b** Cryo-transmission electron microscopy (cryo-TEM) image of a large multilamellar vesicle prepared as per the vesicles used for the microfluidics experiments. Note the presence of abundant internal membranes. Scale bar: 100 nm. **c** Fluorescence microscopy images (Texas Red® channel) corresponding to vesicle growth over time. Scale bar: 10 μm. See Supplementary Movie 3. **d** Fluorescence microscopy images (Texas Red® channel) corresponding to vesicle division. The yellow arrows indicate the formation and departure of a daughter vesicle. Scale bar: 20 μm. Inset depicts the daughter vesicle in the Alexa Fluor® 488 channel. In **c** and **d**, the images are scaled logarithmically to enhance visibility of internal membranous structures. See Supplementary Movie 4. **e** HPLC-MS experiment demonstrating formation of phospholipid **5** (indicated in red arrow) upon addition of lysolipid **4** and other precursors to phospholipid **3** giant vesicles encapsulating FadD10

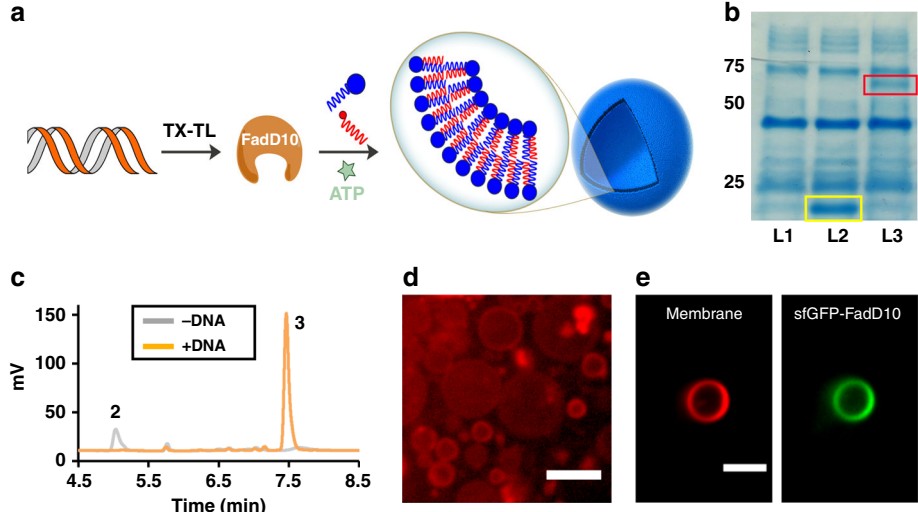

**Fig. 4** Linking gene expression of FadD10 to phospholipid synthesis. **a** Schematic representation of the cell-free expression of FadD10 and subsequent assembly of the de novo synthesized phospholipid into vesicles in the presence of appropriate reactive precursors [TX-TL: transcription/translation]. **b** SDS–PAGE analysis of the expression of FadD10 in the PURExpress® System. Lane L1: No DNA; Lane L2: DHFR DNA; Lane L3: FadD10 DNA. **c** HPLC/ELSD traces monitoring the formation of phospholipid **3** by incubation of PURExpress® System with an aqueous solution of dodecanoic acid, lysolipid **2**, ATP and MgCl₂ at 37 °C in the absence (gray line) or presence (orange line) of plasmid DNA coding for FadD10. **d** Spinning disk confocal microscopy of the in situ formed phospholipid vesicles in the PURExpress® System driven by FadD10 expression. Membranes were stained using 0.1 mol% Texas Red® DHPE dye. Scale bar: 5 μm. **e** Localization of sfGFP-FadD10 to the membrane of the vesicles formed upon addition of the plasmid encoding the former into PURE system. External proteins were digested by Proteinase K. Scale bar: 5 μm

and membrane formation to enable genetic circuits and sensors to effect cell membrane growth. We therefore linked the expression of FadD10 to lipid synthesis using a cell-free protein expression system (Fig. 4a). In a typical reaction, plasmid DNA encoding FadD10 was first added to the PURExpress® System, a minimal recombinant transcription/translation system. We confirmed FadD10 expression by SDS–PAGE (Fig. 4b, Supplementary Fig. 16a). Then, we added lipid precursors and incubated the mixture at 37 °C for 3–6 h. HPLC-ELSD traces of the reaction mixture indicated that phospholipid **3** was formed, and no lysolipid **2** was detectable (Fig. 4c). This result demonstrates that FAAs can be continuously generated in a complex medium, such as PURE system, and subsequently react with an amine-functionalized lysolipid to form the corresponding phospholipid. We observed a large number of vesicles in the reaction mixture using microscopy (Fig. 4d and Supplementary Fig. 13c). When we omitted FadD10 DNA, no phospholipid synthesis occurred (Fig. 4c), and no vesicles were observed (Supplementary Fig. 16b). When we replicated the experiment using a plasmid encoding sfGFP fused to FadD10 (Supplementary Fig. 17a–d for characterization, Supplementary Table 1 for primers), we found that the enzyme was spontaneously associated with the de novo formed membranes (Fig. 4e). These results suggest that expression of FadD10 will enable the coupling of genotype with membrane growth in synthetic cells.

## Discussion

In summary, we have developed an efficient and minimal biochemical route to synthesize phospholipids by repurposing the activity of a single enzyme. Our strategy of using a soluble enzyme to promote phospholipid formation has a distinct advantage over previously described methods using integral membrane proteins[27,31,32]. The latter are difficult to reconstitute and require pre-existing membranes for maintaining proper structure and function. A method that relies on a soluble enzyme could be more straightforward and effective in linking genetic circuits with membrane formation in artificial cells.

Finally, our results also shed light on possible mechanisms leading to the origins of modern cellular membranes. Although it is believed that the earliest protocell membranes were composed of simple single-chain amphiphiles derived from geochemical processes[33], there is still no consensus on how complex lipid synthesis arose during evolution. It has been hypothesized that phospholipid synthesis may have arisen in the RNA-world[4]. If so, then a ribozyme capable of driving phospholipid synthesis would be required[5]. If ribozymes were to closely mimic the function of natural lipid synthesizing acyltransferases, they would most likely be membrane-bound to interact effectively with their lipid substrates. This possibility seems unlikely, since RNAs are highly charged and polar, which makes them unlikely to embed catalytic centers within lipid membranes[34]. Our approach suggests an alternative scenario. Early ribozymes may have evolved to activate single-chain precursors, such as fatty acids into corresponding acyl adenylates. The coupling of such activated precursors to other single-chain amphiphiles may have occurred spontaneously, either in micelles or within membranes. Indeed, a ribozyme capable of synthesizing acyl phosphates has previously been identified[35], and the prebiotic plausibility of a single-chain phosphorylated amino amphiphile has been suggested[36] (see Supplementary Fig. 18 for reaction scheme). If fatty acyl adenylate synthesizing ribozymes could be selected through directed evolution and used to drive the de novo formation of membrane-forming lipids, it would suggest that catalysis by a soluble macromolecule could have played an important role in the early evolution of complex cellular membranes.

The use of soluble enzymes to efficiently generate membrane-forming lipids, de novo could have numerous applications. There is tremendous interest in the development of synthetic cells[37–39], and advanced synthetic cells will undoubtedly require simplified methods to generate and maintain phospholipid membranes[40,41]. Our system provides a means to link gene expression and lipid formation in a synthetic cell. We also envision that the methodology of enzymatic de novo phospholipid membrane formation will enable synthesis of specific lipids in a cellular milieu, by expressing FadD10 and providing the reactive precursors. Finally, our method provides a route to synthesize proteoliposomes, which could have applications in biomaterials design and the reconstitution of disease-relevant membrane-bound proteins[10,11].

## Methods

**HPLC-ELSD-MS analyses.** Solvent mixtures for chromatography are reported as volume/volume (v/v) ratios. HPLC analyses were performed with an Eclipse Plus C8 analytical column with gradients based on Phase A and Phase B (where Phase A: $H_2O$ with 0.1% formic acid; Phase B: MeOH with 0.1% formic acid). For purification purposes, Zorbax SB-C18 semi-preparative column was used with gradients based on Phase A and Phase B gradients (where Phase A: $H_2O$ with 0.1% formic acid; Phase B: MeOH with 0.1% formic acid). Agilent 6230 Accurate-Mass TOFMS mass spectrometer was used to obtain electrospray ionization-time-of-flight (ESI-TOF) mass spectra. A Varian 380-LC detector was used to record ELSD chromatograms.

**NMR spectroscopy.** Proton nuclear magnetic resonance ($^1H$ NMR) spectra were acquired on a Varian VX-500 MHz or Jeol Delta ECA-500 MHz spectrometers, and were referenced relative to residual proton resonances in $CDCl_3$ (at δ 7.24 ppm), $d_6$-DMSO (at δ 2.50 ppm), or $CD_3OD$ (at δ 4.87 or 3.31 ppm). Chemical shifts were reported in parts per million (ppm, δ) relative to tetramethylsilane (δ 0.00). The splitting patterns in $^1H$ NMR are designated as singlet (s), doublet (d), triplet (t), quartet (q), or pentuplet (p). All first-order splitting patterns were designated based on the appearance of the multiplet. Splitting patterns that could not be interpreted readily are assigned as multiplet (m) or broad (br). $^{13}C$ NMR spectra were recorded on a Varian VX-500 MHz or Jeol Delta ECA-500 MHz spectrometers, and were referenced relative to residual proton resonances in $CDCl_3$ (at δ 77.23 ppm), $CD_3OD$ (at δ 49.15 ppm), or $d_6$-DMSO (at δ 39.51 ppm). $^1H$ and $^{13}C$ NMR spectra for compounds **1–6** and various synthetic intermediates (**8–11**) are provided in the Supplementary Figs. 19–28.

**Synthesis of dodecanoyl-AMP (1).** At first, anhydrous $Et_2O$ (7 mL) was used to dissolve solid dodecanoic acid (241.2 mg, 1.20 mmol) and stirred for 10 min at RT. Then, a 1 M solution of DCC in $CH_2Cl_2$ (1.2 mL) previously diluted with anhydrous $Et_2O$ (4 mL) was added dropwise for 10 min. The white suspension that was produced was continuously stirred at RT for 12 h. Afterward, the mixture was filtered, and the filtrate was evaporated in vacuo to afford a white solid. The corresponding dodecanoic anhydride was dried under high vacuum for 3 h and used without further purification.

5′-AMP.$H_2O$ (147.8 mg, 0.40 mmol) was dissolved in a solution of 8 mL of 1:1 $H_2O$:pyridine (v/v) containing NaOH (20 mg, 0.50 mmol) taken in a long vessel. Following this, a solution of dodecanoic anhydride dissolved in 8 mL of THF was added to the mixture in three portions. Upon addition, a white suspension formed readily, and the mixture was stirred at RT continuously. In about 15 min, the suspension turned into nearly clear solution. Next, $Et_2O$ (20 mL) was added and the two-phase mixture was stirred vigorously for about 2 min. Then, the two-phase system was allowed to rest briefly and the upper phase was removed carefully. This process was repeated three times. After this, a small volume of distilled water (~3 mL) was added, and the pH was lowered to ~3 by dropwise addition of aqueous HCl (5%). The gel was transformed into a white suspension and $Et_2O$ (3 × 15 mL) was used to extract the latter by vigorous stirring. The top organic layer is separated continuously after each round. The white solid was transferred to a Büchner funnel and the residual suspension was filtered. Finally, the solid is washed with acetone (3 × 5 mL) and $Et_2O$ (2 × 5 mL). The residue is dried in vacuo, and dodecanoyl-AMP **1** is obtained as a white powder [125.2 mg, 60%]. $^1H$ NMR ($d_6$-DMSO, 500.13 MHz, δ): 8.55 (s, 1 H, 1 × $CH_{Ar}$), 8.30 (s, 1 H, 1 × $CH_{Ar}$), 5.94 (d, J = 5.6 Hz, 1 H, 1 × CH), 4.57 (t, J = 5.3 Hz, 1 H, 1 × CH), 4.24–3.98 (m, 4 H, 2 × CH + 1 × $CH_2$), 2.33 (t, J = 7.3 Hz, 2 H, 1 × $CH_2$), 1.55–1.35 (m, 2 H, 1 × $CH_2$), 1.31–1.09 (m, 16 H, 8 × $CH_2$), 0.88 (t, J = 6.8 Hz, 3 H, 1 × $CH_3$). $^{13}C$ NMR ($d_6$-DMSO, 125.77 MHz, δ): 163.3, 152.8, 148.8, 148.5, 140.8, 118.7, 87.3, 83.6, 73.8, 70.4, 66.0, 34.6, 31.3, 29.0, 29.0, 28.9, 28.7, 28.7, 28.3, 24.1, 22.1, 14.0. MS (ESI-TOF) [m/z (%)]: 530 ($[MH]^+$, 100). HRMS (ESI-TOF) calculated for $C_{22}H_{37}N_5O_8P$ ($[MH]^+$) 530.2374, found 530.2374.

**Expression and purification of His$_6$-tagged FadD10.** The *fadD10* (*Rv0099*)-pDEST17 plasmid was transformed into Novagen BL21(DE3) competent *E. coli*

cells and grown overnight at 37 °C in Luria-Bertani (LB) broth containing 0.1 mg/mL of ampicillin. Afterward, 1 mL of the overnight culture was used to inoculate 1 L of freshly autoclaved LB medium containing 0.1 mg/mL of ampicillin. The rest of the overnight culture was stored as 25% glycerol stocks at −80 °C. The culture was grown at 37 °C in a shaker-incubator till the $OD_{600}$ reached about 0.7. Overexpression of FadD10 was induced by addition of 1 mM isopropyl 1-thio-D-galactopyranoside (IPTG). The cells were then grown for 18–20 h at 18 °C, after which the cells were harvested by centrifuging at 6000 rcf for 20 min at 4 °C. The pellet was resuspended by vortexing in 10 mL of lysis buffer containing 25 mM Tris pH 8.0, 0.5 M NaCl, 2 mM β-mercaptoethanol, 1 mg/mL lysozyme and a cocktail of protease inhibitors (SigmaFast®). Following cell lysis by an ultra-sonicator probe, debris were removed by centrifuging (13,000 rcf, 20 min, 4 °C). The supernatant was incubated in a gravity column with $Ni^{2+}$-nitrilotriacetate (Ni-NTA) agarose resin pre-equilibrated with 10 mM imidazole on a shaker table for 3 h at 4 °C. Next, the flow-through was discarded and the resin was washed with 10 mL each of 20 mM imidazole and 50 mM imidazole solutions. Finally, $His_6$-tagged FadD10 was eluted with 250 mM imidazole and collected in 1 mL fractions and the pure fractions were pooled and concentrated using 30 kDa MWCO centrifuge filters at 4 °C. They were stored in small aliquots containing 10% glycerol each (final concentration: 6.75 mg/mL) and stored at −80 °C.

**De novo phospholipid formation mediated by FadD10**. In a typical de novo phospholipid synthesis reaction, to a 2.0 μL of lysolipid **2** solution (10 mM stock in $H_2O$) were successively added 2 μL HEPES (1 M), 1.8 μL of $MgCl_2$ (100 mM stock solution), 0.6 μL of ATP (100 mM stock solution) and 9.6 μL of $H_2O$. Next, 3.3–13 μg of FadD10 (from a 10% glycerol stock, 6.75 mg/mL) were added, and the solution was mixed by gentle tapping. Afterward, 2.0 μL of dodecanoic acid (as 10 mM sodium dodecanoate stock solution) were added. The reaction mixture was kept incubated in a 37 °C water bath. Small aliquots (~2 μL) were taken out at various time points and placed on a glass slide for microscopic observations. We found that the pH range 7.5–8.5 is optimum for vesicle formation. Below pH 7.5, vesicle formation was impaired due to aggregation caused by poor solubility of dodecanoic acid. Above pH 8.5, the amino-lysolipids and phospholipids had reduced stability due to hydrolysis at the *sn2* position. We carried out in situ vesicle formation experiments in the concentration range 0.1–1.0 mM of the amphiphilic precursors (see Supplementary Fig. 8 for representative images).

**FRET assay for de novo phospholipid membrane formation**. A phospholipid synthesis reaction was setup using 0.5 mM lysolipid **2**, 0.5 mM sodium dodecanoate, 7.5 mM $MgCl_2$, 2.5 mM ATP, and FadD10 in HEPES buffer (pH 8.0, 100 mM) in presence of 0.5 μM of each of the FRET dyes NBD-DHPE (donor; $\lambda_{em}$: 530 nm) and Rhodamine-DHPE (acceptor; $\lambda_{em}$: 586 nm). The reaction mixture was placed in a 384-well plate at 37 °C. Four hundred and thirty nanometers (bandwidth: 7.5 nm) were used as the wavelength for excitation, and emission was monitored at 530 nm (bandwidth: 7.5 nm) and 586 nm (bandwidth: 7.5 nm). The ratio of the fluorescence intensities at 530 nm and 586 nm was calculated and plotted with time. In a control experiment, ATP was substituted with GTP and fluorescence measurements were carried out using same parameters.

**Microfluidics experiments**. Giant vesicles were prepared by adding the following components in the given order: 20 μL lysolipid **2** (10 mM in 50 mM HEPES pH 7.5 buffer), 9 μL $MgCl_2$ (100 mM), 3 μL ATP (100 mM), 20 μL ddH₂O, 20 μL 100 mM HEPES pH 7.5 buffer, 6.5 μL FadD10 (6.75 mg/mL), 1.5 μL Alexa Fluor® 488-labeled FadD10 (14 μM), 0.5 μL Texas Red® DHPE (100 μM in EtOH), and 20 μL FAA **1** (10 mM in 50 mM HEPES pH 7.5 buffer). The reaction mixture was placed in a glass vial and stirred at RT for 2 h, obtaining phospholipid **3** vesicles encapsulating FadD10. We also carried out vesicle formation following the described procedure in presence of 0.1 mM TNP-ATP, a surrogate of ATP that binds to ATP-binding proteins resulting in marked increase in fluorescence intensity. We observed that the fluorescence intensity is significantly higher inside the vesicles compared with the background (Supplementary Fig. 13f), suggesting that FadD10 is encapsulated highly efficiently in functional form.

Long-term vesicle growth was observed in a microfluidic device fabricated from polydimethylsiloxane (PDMS) and plasma bonded to cover glass. Pre-formed vesicles were vacuum-loaded into chambers that were 230-μm wide, 230-μm long, and 80-μm high. The chip was then connected to a reservoir of precursor solution (40 μM lysolipid **4**, 40 μM sodium dodecanoate, 3 mM ATP, 9 mM $MgCl_2$, 40 mM HEPES pH 7.5 buffer) and placed in a temperature-controlled microscope box at 37 °C. In each experiment, the osmolarities of the vesicle solution and the flow solution were measured and adjusted by adding glycerol accordingly prior to loading using an osmometer. Precursor solution flow was started immediately after the first image of the time-lapse Supplementary Movies were acquired. Flow was regulated to ~20 μL/h using air pressure. Precursors flowing in the main channel of the device entered vesicle chambers by diffusion through connections that were 50-μm wide, 60-μm long, and 50-μm high. This architecture allows observation of vesicles under shear-free conditions without convective flow while ensuring a constant supply of precursors, which are expected to completely equilibrate within 10 min. Alternatively, in the case of control experiments, lysolipid **4** was replaced by 1-palmitoyl-2-hydroxy-*sn*-glycero-3-phosphocholine (Lyso $C_{16:0}$ PC-OH).

**De novo phospholipid formation in PURExpress® System**. In a typical 10 μL protein expression reaction, the following components were added in the given order: 4 μL of Solution A (containing amino acids, energy factors, etc.), 3 μL of Solution B (containing ribosomes, aminoacyl tRNA synthetases, etc), 0.2 μL (4 U) murine RNase inhibitor (New England Biolabs), *x* μL nuclease-free $H_2O$, *y* μL DNA. Initially, protein expression was optimized using different amounts of FadD10 DNA: 50 ng, 100 ng, 150 ng, and 200 ng. A negative control (no DNA) and a positive control (50 ng DHFR DNA) experiment was also carried out at the same time. All reaction mixtures were incubated in a 37 °C water bath for 3 h and then analyzed by SDS–PAGE. In total, 2.5 μL of each reaction mixture was loaded onto a precast 4–20% polyacrylamide gel (Mini Protean, Bio-Rad). The gel was run for 50 min at 130 V and then stained with Coomassie Blue (Instant Blue®, CBS Scientific) for 1 h. The stained gel was washed with distilled water and imaged with a commercial scanner. Expression of FadD10 was observed as expected. It was found that 50 ng DNA was sufficient for a 5 μL protein expression reaction for FadD10. Increasing the amount of DNA did not lead to any significant change in the protein expression (Supplementary Fig. 16a). Expression of sfGFP-FadD10 was carried out in a similar manner and monitored using spectro-fluorimetrically (Supplementary Fig. 17b).

For phospholipid synthesis reactions, FadD10 was first expressed in PURExpress® System with the total reaction volume as 10 μL. A mixture containing precursors for lipid synthesis was prepared by adding 1.25 μL lysolipid **2** (5 mM), 0.25 μL sodium dodecanoate (25 mM), 1.0 μL ATP (100 mM), 2.0 μL $MgCl_2$ (100 mM), and 10.5 μL 100 mM HEPES pH 8.0 buffer. Then, it was added to the PURExpress® System solution, and the resulting reaction mixture was further incubated at 37 °C for 3 h with tumbling. Afterward, 25 μL of MeOH was added to the reaction mixture, which cause formation of a white precipitate. The precipitate was separated by brief centrifugation, and the supernatant was collected and injected into HPLC-ELSD-MS. Formation of phospholipid **3** was confirmed from the retention time and verified by mass spectrometry. In separate experiments, it was also found that higher concentrations of phospholipid **3** (tested up to 1 mM) can be synthesized, albeit with longer incubation times (up to 12 h). Alternatively, in a control experiment, where FadD10 DNA was omitted in the first step, no phospholipid formation was observed.

Synthesis of phospholipid **3** (in the concentration range 0.25–1.0 mM) was carried out in PURExpress® System as described before using FadD10 and sfGFP-FadD10 plasmids. Vesicle formation can be improved and aggregate formation can be minimized by tumbling, occasional mild vortexing, and tapping the reaction tube. Longer incubation time (24 h) was required when lipid synthesis (0.5 mM) was carried out using sfGFP-FadD10 plasmid. The membranes were stained using 0.1 mol% Texas Red® DHPE. An aliquot of the corresponding sample was placed on a glass slide and observed using fluorescence microscopy. In a control experiment where no DNA was added in the beginning, vesicles were not observed.

**Cryogenic-transmission electron microscopy (Cryo-TEM)**. Copper grids (Quantifoil R2/2, 300 mesh copper; 2 μm hole/2-μm spacing) were treated in the beginning with $CHCl_3$ for 4 h to get rid of residual plastic. Later, they were dried, and their surfaces were glow discharged at 20 mA for 30 s using an Emitech K950X instrument with a K350 glow discharge unit. Once the surface for vesicle adhesion is ready, samples were vitrified by plunge-freezing using the commercial environmentally controlled automated Vitrobot (FEI, The Netherlands) at a controlled temperature of 4 °C and ≈95% humidity [Conditions Vitrobot: 1 blot, with 4 s blot time using standard Vitrobot filter paper, φ55/20 mm, Grade 595; Blot force: 0, Drain time: 0][42]. A 3.5 μL of in situ formed phospholipid **3** vesicles was deposited on the grid surface. This solution was allowed to sit for 4 s. After this, the grids were blotted on both sides using Teflon sheets from reverse and on the backside by using filter paper. Next, they were plunged into the liquid ethane immediately at the temperature of liquid nitrogen. On imaging by cryo-TEM, spherical compartments were detected in accordance with the vesicle architecture.

**Reporting Summary**. Further information on experimental design is available in the Nature Research Reporting Summary linked to this Article.

## Data availability
The authors declare that the data supporting the findings of this study are available within the paper and its Supplementary Information files, and, also available from the corresponding author upon reasonable request.

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

## Acknowledgements

This material is based upon work partially supported by the Department of Defense (Army Research Office) through the Multidisciplinary University Research Initiative (MURI) under Award No. W911NF-13-1-0383 and a Young Investigator Grant from the Human Frontier Science Research Program (RGY0066/2017). R.J.B. thanks the Human Frontier Science Program (HFSP) for his Cross-Disciplinary Fellowship. H.N. is supported by a Swiss National Science Foundation fellowship. We sincerely thank Prof. James Sacchettini (Texas A&M University) for generously providing us the plasmid for FadD10. We also thank Prof. Jeff Hasty, Andriy Didovyk, Omar Din, and Ryan Johnson at UC San Diego for their collaboration with the microfluidic chip fabrication and experimental setup.

## Author contributions

A.B., R.J.B., H.N., and N.K.D. conceived and designed the experiments. A.B., R.J.B., and H.N. performed the experiments. A.B., R.J.B., H.N., and N.K.D. analyzed the data. A.B., R.J.B., H.N., and N.K.D. co-wrote the paper.

## Additional information

**Competing interests:** The authors declare no competing interests.

