## [Peer Review File · Nature Communications]

Reviewers' comments:

Reviewer #1 (Remarks to the Author):

In this manuscript entitled as “a minimal biochemical route to synthetic phospholipids membrane”, Bhattacharya et al. have signified the existence of minimal route which utilizes mainly of the soluble protein for membrane phospholipids synthesis. Therein, they have shown a simplified pathway for the synthesis of phospholipids starting from fatty acid, amine activated lysolipid and ATP. This has been illustrated to result in the de novo formation of membrane. Additionally, confocal microscopy images and movie data show the growth, shape deformation and subsequently vesicle division resulted from the encapsulated soluble protein FadD10, with the required precursors supplied from the outside. Finally, the authors have also functionally recapitulated the phospholipids membrane synthesizing route using cell-free synthesized FadD10. With all this, the authors appear to prove their hypothesis. The reviewer found the study as interesting and conceptually strong. Having saying this, the reviewer would like to raise the following points for the improvement of the current state of the manuscript.

1. As the novel point in the manuscript is the de novo synthesis of membrane phospholipid, and eventually vesicle, slight modification would like to improve the title. For example, “a minimal biochemical route towards the de novo formation of synthetic phospholipids membrane”.
2. Figure 2c: the normalized value given in the Y-axis makes the plot difficult to make a direct correspondence between formation of phospholipids 3 and consumption of lysolipid 2. Hence, it is better to replace the Y-axis value with absolute one.
3. Figure 3b: the authors used the phrase “giant vesicle” for the cryo-TEM image. Is it really giant vesicle? It is better to replace it with large multilamellar vesicle instead.
4. Supplementary figure 5a shows the instability of DDA in the presence of 10 mM of Mg²⁺. The authors need to provide adequate justification on how the stability of DDA was maintained in the scenario where FadD10 was synthesized in the PURE system (in view of the presence of higher concentration of Mg²⁺).
5. Supplementary figure 5d and 5e: Please provide the concentration of 1 (FAA) and lysolipid 2 used in the chemical competition experiment.
7. Supplementary figure 9: it looks important to add one more negative control. Such as, to check the membrane localization of FadD10 in the presence of ATP and Mg²⁺, and only in the absence of DDA.
8. Supplementary figure 9c: In how many vesicles have you observed such membrane localization of FadD10? A population analysis is required.

9. Supplementary figure 10e: why do the authors need to include such data for the characterization of the vesicles used in the microfluidic experiment? If it is to check the permeability of the membrane for smaller water-soluble molecules such as ATP, as they were supplemented externally, it is much better to directly probe the level of permeability of the membrane for ATP. For this, the authors can use a fluorescent ATP probe like ATeam or other.

Reviewer #2 (Remarks to the Author):

This manuscript by Devaraj and coworkers describes a novel lipid and liposome synthesis technique using lysolipid and fatty acid chain precursors catalyzed by a repurposed soluble enzyme. It is the first study of this kind as far as I can tell and in my opinion quite creative. The authors first show that fatty acid adenylates (FAAs) can react with amine-functionalized lysolipids to form phospholipids. They then convincingly demonstrate that FadD10, a recently characterized fatty acid adenylate ligase, can convert a long chain fatty acid to a FAA which can then react with amine functionalized lysolipids to form phospholipids that form vesicles within which FadD10 is encapsulated and continues to produce more phospholipids.

Overall the work is elegant and poses an important hypothesis about the origin of the lipid bilayer membrane that does not depend on a presence of a membrane and membrane bound membrane protein. A few things that I think still need further clarification and discuss include

1. The authors use amine functionlized lysolipids. While soluble enzymes and fatty acids can be imagined in the origin of membrane materials, the presence of such specifically functionalized lipids is a barrier to this system to be considered as a model for origin of lipid membranes in organisms. Perhaps the authors can expand on the nature of the proposed prebiotic phosphorylated amino amphiphile and how it can be converted into lipids?
2. Is there a primitive analog of the FadD10 protein that the authors can point to that may be responsible for the first lipid membrane synthesis?
3. In the materials of methods section/Cryo TEM subsection – “flunge-freezing” should be “plunge-freezing”

Reviewer #3 (Remarks to the Author):

Bhattacharya and colleagues present a new method for de novo vesicle formation from activated fatty acids that react spontaneously with amino-functionalized lysolipids to form phospholipids. A water-soluble enzyme is used to convert fatty acids into fatty acyl adenylates, and the approach to making phospholipids is very different from how the molecules are made in living cells. The work appears well done but the research is somewhat preliminary. A more complete paper would present a valuable addition to the emerging field of true bottom-up synthetic biology.

Comments:

1. The authors attribute the high selectivity of their synthesis to the hydrophobic interactions between activated fatty acids and amino-functionalized lysolipids. Do the fatty acyl adenylates react with amines on hydrophobic proteins? The application of the system lies in combining the synthesis of lipids and vesicle growth with the incorporation of (membrane) proteins. Moreover, the enzyme FAD10 is naturally involved in the biosynthesis of lipopeptides, and I would thus expect similar byproducts in the presence of proteins and peptides. How is or can this be prevented?
2. What is the range of reaction conditions useful for vesicle formation (concentrations of reactants, molar excess)? And what is the scale (amount) on which the activated fatty acids can be formed. Why is Mg^{2+} present in 3-fold excess over ATP?
3. The vesicle formation is not well characterized: (a) how much of FAD10 does end up on either side of the membrane? (b) FAD10 is described to be water-soluble but why then is it primarily found associated with the membrane? (c) where does the phospholipid synthesis occur (all over the membrane or from discrete spots)? (d) the process of vesicle growth is poorly described (although Suppl. Movies S3 and S4 seem to present some clues as to what is happening).
4. It is unlikely that phospholipid synthesis is fully symmetrical, but what then makes the lipids flop-flip from one leaflet to the other?
5. What is the final amphiphile composition of the vesicles? What is the fraction of phospholipids, other precursors? According to Figure 2C, all lysolipid is converted into phospholipid but how much of the other precursor(s) are found in the vesicles or are they only present in the aqueous solution (which I doubt)?
6. What is the ion and solute permeability of the vesicles? In order for the vesicles to grow, lipids must be synthesized on the out- and inside (see also comment 4). I assume that some FAD10 and ATP is trapped in the initial small vesicles, but then how do they grow as the enzyme gets diluted and ATP is hydrolyzed? Does ATP leak into the vesicles? If so, then the membranes are highly permeable to small ions and solutes and of limited use for further studies or applications. If the membranes are impermeable to ATP, then I don't see how they can grow. How sensitive is FAD10 for ADP inhibition?
7. The approach seems rather straightforward and it would thus be valuable to also present the synthesis of lipids with PE, PG and PS headgroups, and form vesicles with non-bilayer zwitterionic

and bilayer-forming anionic lipids in addition to just PC. Most membrane proteins require either or both of these types of lipids for their function.

8. What is the yield of the vesicles? What will ultimately limit the use of the synthesis approach or is there no limit; the reactions are carried out in very small volumes (<20 µL) and the amount of vesicles formed seems low, if I am not mistaken?

9. The authors indicate in the discussion that so far phospholipid biosynthesis in vitro using integral membrane proteins has met limited success. However, I readily found the following recent papers:

Cell-Free Phospholipid Biosynthesis by Gene-Encoded Enzymes Reconstituted in Liposomes.

Scott A, Noga MJ, de Graaf P, Westerlaken I, Yildirim E, Danelon C. PLoS One. 2016 Oct 6;11(10):e0163058. doi: 10.1371/journal.pone.0163058

Growing Membranes In Vitro by Continuous Phospholipid Biosynthesis from Free Fatty Acids.

Exterkate M, Caforio A, Stuart MCA, Driessen AJM.

ACS Synth Biol. 2018 Jan 19;7(1):153-165. doi: 10.1021/acssynbio.7b00265.

Responses to the Reviewers Comments:

Responses to Reviewer 1

Comment 1. As the novel point in the manuscript is the de novo synthesis of membrane phospholipid, and eventually vesicle, slight modification would like to improve the title. For example, “a minimal biochemical route towards the de novo formation of synthetic phospholipids membrane”.

Response 1: We thank the reviewer for the suggestion. We have modified the title accordingly.

Comment 2. Figure 2c: the normalized value given in the Y-axis makes the plot difficult to make a direct correspondence between formation of phospholipids 3 and consumption of lysolipid 2. Hence, it is better to replace the Y-axis value with absolute one.

Response 2: We thank the reviewer for the suggestion. We have modified the plot with the absolute values of the HPLC areas of the lysolipid 2 and phospholipid 3 in the revised manuscript.

Comment 3. Figure 3b: the authors used the phrase “giant vesicle” for the cryo-TEM image. Is it really giant vesicle? It is better to replace it with large multilamellar vesicle instead.

Response 3: We thank the reviewer for the comment. We have made the change as suggested.

Comment 4. Supplementary figure 5a shows the instability of DDA in the presence of 10 mM of Mg^{2+} . The authors need to provide adequate justification on how the stability of DDA was maintained in the scenario where FadD10 was synthesized in the PURE system (in view of the presence of higher concentration of Mg^{2+}).

Response 4: We thank the reviewer for the comment and would like to clarify this point. The dodecanoyl-AMP intermediate is generated by the enzyme FadD10 from dodecanoic acid and ATP and it reacts rapidly with the amino-lysolipid 2 to form the phospholipid 3. We determined the 2nd order rate constant for this step to be $87.0 \pm 9.1 \text{ M}^{-1} \text{ s}^{-1}$. So, the life-time of the FAA intermediate is very small. So, even if there is any hydrolysis by Mg^{2+} , it will be negligible. For the sake of argument, even if we consider that there is some hydrolysis of the FAA, we have provided excess ATP (3 times by molar ratio) over the dodecanoic acid. So, any dodecanoic acid obtained from the hydrolysis of FAA can be replenished. We have added a sentence in the revised manuscript to highlight the fact that FAAs can be continuously generated in the PURE system and rapidly converted to phospholipid upon reaction with an amino-lysolipid.

Comment 5. Supplementary figure 5d and 5e: Please provide the concentration of 1 (FAA) and lysolipid 2 used in the chemical competition experiment.

Response 5: We have provided the concentrations of FAA 1 and lysolipid 2 in Supplementary Fig. 5d and 5e as the reviewer suggested.

Comment 6. Supplementary figure 9: it looks important to add one more negative control. Such as, to check the membrane localization of FadD10 in the presence of ATP and Mg^{2+} , and only in the absence of DDA.

Response 6: We performed this control experiment. If DDA is omitted, we found that the amino-lysolipid 4 (at the same concentration as that used for the described experiment) causes the encapsulated FadD10 to leak out rapidly, likely due to formation of pores. We included this data in Supplementary Fig. 12 in the revised manuscript.

Comment 7. Supplementary figure 9c: In how many vesicles have you observed such membrane localization of FadD10? A population analysis is required.

Response 7: We thank the reviewer for this comment. We obtained membrane localization of labeled FadD10 in all ($n = 49$) of the GUVs observed in three experiments. We have added this data in the supplementary information section. We have also updated Supplementary Fig. 12 (previously Supplementary Fig. 9) and included images showing multiple GUVs with Alexa Fluor 488 labeled FadD10 localized to the membrane upon addition of amphiphilic precursors.

Comment 8. Supplementary figure 10e: why do the authors need to include such data for the characterization of the vesicles used in the microfluidic experiment? If it is to check the permeability of the membrane for smaller water-soluble molecules such as ATP, as they were supplemented externally, it is much better to directly probe the level of permeability of the membrane for ATP. For this, the authors can use a fluorescent ATP probe like ATeam or other.

Response 8: We thank the reviewer for the comment. Regarding supplementary figure 10e, our intent was to show that the vesicles can retain charged water-soluble molecules such as HPTS. In fact, we had shown that TNP-ATP, a fluorescent analogue of ATP, can also be retained in the vesicles. This data was present in Supplementary Fig. 10f in our initial submission (Supplementary Fig. 13f in the revised manuscript).

Responses to Reviewer 2

Comment 1. The authors use amine functionalized lysolipids. While soluble enzymes and fatty acids can be imagined in the origin of membrane materials, the presence of such specifically functionalized lipids is a barrier to this system to be considered as a model for origin of lipid membranes in organisms. Perhaps the authors can expand on the nature of the proposed prebiotic phosphorylated amino amphiphile and how it can be converted into lipids?

Response 1: We thank the reviewer for asking us to clarify this point. Sutherland and coworkers had previously proposed a prebiotically plausible route to single-chain phosphorylated amino-amphiphiles (*Angew. Chem. Int. Ed.* **2007**, *46*, 4166-4168). We have been able to replicate their work in our lab. We propose that, a fatty acyl adenylate will be capable of reacting with the phosphorylated amino-amphiphile to form a two-chain phospholipid-like molecule. We propose the following scheme for such a transformation and have now included the proposed scheme in the supplementary information section as an aid to the reader (Supplementary Fig. 18):

Comment 2. Is there a primitive analog of the FadD10 protein that the authors can point to that may be responsible for the first lipid membrane synthesis?

Response 2: As of present, a primitive analogue of FadD10, that can activate fatty acids is not known. However, given the diversity of acyl adenylate forming enzymes in all domains of life, and ubiquitous need for carboxylic acid activation, one may be discovered in the future. In fact, a ribozyme capable of activating small carboxylic acids into their corresponding acyl phosphates has been described by Yarus *et al.* (*Biochemistry*, **2001**, 40, 6998-7004). It may be possible to select a fatty acyl adenylating ribozyme through *in vitro* directed evolution.

Comment 3. In the materials of methods section/Cryo TEM subsection – “flunge-freezing” should be “plunge-freezing”

Response 3: We have corrected the typo in the revised manuscript.

Responses to Reviewer 3

Comment 1. The authors attribute the high selectivity of their synthesis to the hydrophobic interactions between activated fatty acids and amino-functionalized lysolipids. Do the fatty acyl adenylates react with amines on hydrophobic proteins? The application of the system lies in combining the synthesis of lipids and vesicle growth with the incorporation of (membrane) proteins. Moreover, the enzyme FAD10 is naturally involved in the biosynthesis of lipopeptides, and I would thus expect similar byproducts in the presence of proteins and peptides. How is or can this be prevented?

Response 1: We agree with the reviewer that there is a likelihood that fatty acyl adenylates can react with the amine side groups of hydrophobic membrane proteins. We have not tested if the FAAs can acylate the side chains of any membrane protein. However, we reason that membrane proteins diffuse much slower than amino-lysolipids within a membrane. Also, it will be likely for practical purposes that a molar excess of the amino-lysolipid will be used compared to the membrane proteins. So, the possibilities of encounter of a FAA with an amino-lysolipid molecule will be higher compared to that with a membrane protein. In *Mycobacterium tuberculosis*, FadD10 is a part of a non-ribosomal peptide synthase (NRPS) that is involved in the synthesis of a lipopeptide virulence factor (*Chem. Biol.*, **2007**, 14, 543-551). It has also been shown that FadD10 catalyzes the activation of fatty acids to corresponding adenylates and their subsequent transfer to an acyl carrier protein (encoded by the gene *Rv0100*) to produce an acyl-ACP (*J. Biol. Chem.*, **2013**, 288, 18473-18483). It is this acyl-ACP which is the precursor for the lipopeptide. There is no evidence in the literature that suggests that the biosynthesis of the lipopeptides takes place via non-enzymatic coupling with fatty acyl adenylates.

Comment 2. What is the range of reaction conditions useful for vesicle formation (concentrations of reactants, molar excess)? And what is the scale (amount) on which the activated fatty acids can be formed. Why is Mg²⁺ present in 3-fold excess over ATP?

Response 2: We would like to point out to the reviewer that we had provided the details of the conditions (pH, amphiphile concentrations) in the methods section of the manuscript in our initial submission. To clarify the influence of reaction conditions to our readers, we have further added a set of representative images of vesicle formation from various concentrations of the amphiphilic precursors (Supplementary Fig. 8). We did not observe any accumulation of the fatty acyl adenylates on course of the reaction. This is likely because the non-enzymatic coupling of the FAAs with the amino-lysolipids is extremely rapid. In absence of the amino-lysolipid, there is no significant accumulation of the FAA. This is likely because the FAA can undergo hydrolysis to the corresponding fatty acid and AMP catalyzed by Mg²⁺ ions. We maintained a 3:1 stoichiometry

of Mg^{2+} and ATP based on the previous work by Liu *et al.* (*J. Biol. Chem.* **2013**, 288, 18473-18483) . Initially, we tried carrying out the reaction with 2:1 ratio of Mg^{2+} and ATP and found that the reaction rate is improved when the ratio is changed to 3:1. A likely explanation is that Mg^{2+} coordinates to the acyl phosphate functional group and facilitates the nucleophilic attack by an amine through charge shielding.

Comment 3. The vesicle formation is not well characterized: (a) how much of FAD10 does end up on either side of the membrane? (b) FAD10 is described to be water-soluble but why then is it primarily found associated with the membrane? (c) where does the phospholipid synthesis occur (all over the membrane or from discrete spots)? (d) the process of vesicle growth is poorly described (although Suppl. Movies S3 and S4 seem to present some clues as to what is happening).

Response 3: (a) We would expect a statistically even distribution of FadD10 on both sides of the membrane. (b) We believe that the membrane localization of FadD10 is due to electrostatic interaction with the amphiphilic precursors. Since the theoretical isoelectric point (pI) of His6- FadD10 is 5.4, it will bear an overall negative charge at the pH range we used for phospholipid formation. Under these conditions, it is possible that the enzyme interacts with the amino-lysolipid to be associated with the membranes. (c) We reason that the coupling between the FAA and amino-lysolipid leading to phospholipid synthesis takes place in the membrane due to the amphiphilic nature of the reactants. (d) We believe that the vesicle growth is arising from synthesis of new phospholipid which leads to increase in the membrane surface area. To further characterize vesicle growth, we carried out a previously described FRET-based assay to follow membrane growth during *de novo* phospholipid formation (Supplementary Fig. 9).

Comment 4. It is unlikely that phospholipid synthesis is fully symmetrical, but what then makes the lipids flop-flip from one leaflet to the other?

Response 4: The vesicles are continuously being supplied with the amphiphilic precursors such as amino-lysolipid and dodecanoic acid, which can build-up in the membrane. It has been previously shown that build-up of amphiphilic molecules in the membrane can promote trans-bilayer lipid transfer (Zhelev, *Biophys. J.* **1996**, 71, 257-273). In addition, it has been suggested in previous works with model phospholipid membranes that various amphiphiles can accelerate trans-bilayer lipid transfer (*Biophys. J.*, **2001**, 81, 184-195; *Langmuir*, **2010**, 26, 7307-7313; *FEBS Lett.*, **2010**, 584, 1779-1786).

Comment 5. What is the final amphiphile composition of the vesicles? What is the fraction of phospholipids, other precursors? According to Figure 2C, all lysolipid is converted into phospholipid but how much of the other precursor(s) are found in the vesicles or are they only present in the aqueous solution (which I doubt)?

Response 5: Based on the kinetic studies, the lysolipid was nearly quantitatively consumed during the course of the reaction. If traces of the amphiphilic precursors are remaining, we would expect them to remain preferentially partitioned into the vesicle membranes. In the case of the non-amphiphilic precursors (ATP and MgCl_2) we expect an even distribution between the inside and outside of the vesicles.

Comment 6. What is the ion and solute permeability of the vesicles? In order for the vesicles to grow, lipids must be synthesized on the out- and inside (see also comment 4). I assume that some FAD10 and ATP is trapped in the initial small vesicles, but then how do they grow as the enzyme gets diluted and ATP is hydrolyzed? Does ATP leak into the vesicles? If so, then the membranes are highly permeable to small ions and solutes and of limited use for further studies or applications. If the membranes are impermeable to ATP, then I don't see how they can grow. How sensitive is FAD10 for ADP inhibition?

Response 6: Addition of amphiphilic precursors (lysolipid and fatty acids) is likely to cause a significant change in the phase behavior of the vesicle membranes. It is possible that they are causing local non-lamellar phases

and transient defects in the membranes (Zhelev, *Biophys. J.* **1996**, *71*, 257-273; Burack *et al.*, *Biochemistry*, **1997**, *34*, 10551-10557) and facilitating increased permeability to polar solutes. Transient defects are hypothesized to have significant enhancement effect on transport of solutes and ions as well. Membranes containing both single- and double-chain amphiphiles have been hypothesized to allow transport of charged molecules (ex: NTPs) by transient defect formation (Monnard and Deamer, *Orig. Life Evol. Biospheres* **2001**, *31*, 147-155). Membrane defects are hypothesized to allow passage of molecules as large as tRNAs (plus NTPs and small factors) by Danelon and coworkers (Nourian *et al.*, *Angew. Chem. Int. Ed.* **2012**, *51*, 3114-3118; Scott *et al.*, *PLOS One*, **2016**, *11*, e0163058). Very recently, it has been shown that vesicles composed of a mixture of fatty acids and phospholipids are significantly more permeable to nucleotides compared to pure phospholipid vesicles (Jin *et al.*, *Small* **2018**, *14*, e1704077). So, it is reasonable for us to assume that in presence of the amphiphiles, the membranes will be permeable to ions and solutes. We also reason that enhanced permeability to ions and solutes may be put to advantage in the future for the transportation of molecules across the membrane without the necessity of specialized membrane transporter proteins.

Inhibition of FadD10 by ADP has not been determined by us or not been reported by any other group yet. Also, we would like to point out that ADP is not present in our system. AMP is produced as the by-product of the reaction.

Comment 7. The approach seems rather straightforward and it would thus be valuable to also present the synthesis of lipids with PE, PG and PS headgroups, and form vesicles with non-bilayer zwitterionic and bilayer-forming anionic lipids in addition to just PC. Most membrane proteins require either or both of these types of lipids for their function.

Response 7: We thank the reviewer for the suggestion. As a representative example, we carried out the synthesis of an amine-functionalized lysophosphatidylglycerol (**6**) and performed FadD10-assisted phospholipid synthesis. We obtained the desired bilayer forming anionic phospholipid product (**7**), which self-assembled to form vesicles *in situ*. We have added a brief description of this experiment in the main manuscript and provided additional data for the synthesis and characterization of the corresponding lysolipid and phospholipid in the Supplementary Information section (including Supplementary Fig. 11). We believe that our approach can also be extended to the synthesis of alternative non-bilayer forming phospholipids as well, as long as the lysolipid precursor is modified with a primary amine group.

Comment 8. What is the yield of the vesicles? What will ultimately limit the use of the synthesis approach or is there no limit; the reactions are carried out in very small volumes (<20 microL) and the amount of vesicles formed seems low, if I am not mistaken?

Response 8: We are not clear about what the reviewer meant by “yield of the vesicles”. We have determined the yield of the phospholipids produced in the course of the reaction and provided the results in Fig. 2c. For the revised manuscript, we also carried out a well-established FRET based assay to show that membrane growth took place during *de novo* phospholipid formation (Supplementary Fig. 9). The volume at which the reactions can be carried out is not a limitation. The precursors are inexpensive and the enzyme can be expressed and purified from *E. coli* conveniently. The vesicle images that we have shown are obtained by optical microscopy. If the vesicles that were formed were smaller than the optical resolution limit, they could not be detected by optical microscopy. Indeed, we observed formation of vesicles having size of the order of 100 nm by cryogenic TEM.

Comment 9. The authors indicate in the discussion that so far phospholipid biosynthesis *in vitro* using integral membrane proteins has met limited success. However, I readily found the following recent papers:

Cell-Free Phospholipid Biosynthesis by Gene-Encoded Enzymes Reconstituted in Liposomes. Scott A, Noga MJ, de Graaf P, Westerlaken I, Yildirim E, Danelon C. PLoS One. 2016 Oct 6;11(10):e0163058. doi: 10.1371/journal.pone.0163058

Growing Membranes In Vitro by Continuous Phospholipid Biosynthesis from Free Fatty Acids. Exterkate M, Caforio A, Stuart MCA, Driessen AJM. ACS Synth Biol. 2018 Jan 19;7(1):153-165. doi: 10.1021/acssynbio.7b00265.

Response 9: We thank the reviewer for the comment. We initially wrote that the existing methods of cell free phospholipid synthesis requires integral membrane proteins and pre-existing membranes and the former may be challenging to purify and reconstitute in functional form. Furthermore, the previous approaches relied on membrane proteins, which themselves require pre-existing membranes for proper folding. So, in that sense, true *de novo* phospholipid membrane was not synthesized. In our present work, we have described an alternate route where phospholipids can be synthesized using a soluble enzyme. Also, we have demonstrated that the yield of phospholipids is very high (near quantitative), which is a significant improvement from the previously described methods. However, to avoid any confusion, we have re-written the section and cited the mentioned papers as the reviewer had suggested.

REVIEWERS' COMMENTS:

Reviewer #1 (Remarks to the Author):

The authors have well responded to the points arisen by reviewers and the manuscript is now acceptable for the publication.

Reviewer #2 (Remarks to the Author):

I am satisfied with the responses of the authors to reviewer comments and the changes that they have made to the manuscript. I would be happy to recommend acceptance of the manuscript in its current form.

Reviewer #3 (Remarks to the Author):

The authors have done an excellent job in addressing the points of the reviewers. Although the rebuttal may be available on line, I would include some of the main points of the rebuttal in the paper. E.g . the response given under comments 3, 4 and 6 of reviewer 3 may be relevant for readers of the article and raise further thinking in the field. Their main arguments could easily be incorporated in the discussion section, and in my view it would strengthen the paper even further.

Response to reviewers' comments

Response to Reviewer 1:

Comment: The authors have well responded to the points arisen by reviewers and the manuscript is now acceptable for the publication.

Response: We thank the reviewer for reviewing our work.

Response to Reviewer 2:

Comment: I am satisfied with the responses of the authors to reviewer comments and the changes that they have made to the manuscript. I would be happy to recommend acceptance of the manuscript in its current form.

Response: We thank the reviewer for reviewing our work.

Response to Reviewer 3:

Comment: The authors have done an excellent job in addressing the points of the reviewers. Although the rebuttal may be available online, I would include some of the main points of the rebuttal in the paper. E.g . the response given under comments 3, 4 and 6 of reviewer 3 may be relevant for readers of the article and raise further thinking in the field. Their main arguments could easily be incorporated in the discussion section, and in my view it would strengthen the paper even further.

Response: We thank the reviewer for reviewing our work and providing excellent suggestions. We have incorporated some of the mentioned points in our manuscript.